# Analogy and Visual Content: The *Logica memorativa* of Thomas Murner

**Juan Manuel Campos Benítez**

Faculty of Philosophy and Letters, Meritorious Autonomous University of Puebla, Puebla 72000, Mexico; juan.campos@correo.buap.mx

**Abstract:** In this article, after some thoughts on medieval logic and teaching, we present Thomas Murner's text, *Logica memorativa*, showing some of his mnemonic strategies for the student to learn logic quickly. Murner offers a type of "flash cards" that illustrate much of the teaching of logic at the beginning of the sixteenth century. The first impression is visual, because the cards do not contain words that illustrate their content. Murner's exposition rests on analogies between logic themes that are explained and the visual images presented.

**Keywords:** Murner; medieval logic; memory; analogy

---

## 1. Introduction

Medieval logic pays close attention to language and its uses. We express our thoughts about reality through language and language is the intermediary, so to speak, among human beings. Language is composed of sentences and sentences in turn are composed of words and terms. Terms are classified in different ways: singular, common, indefinite, and quantified terms. This classification is based on its quantity. Another classification gives us univocal, equivocal, and analogical terms.

Medieval logic is complex, it proceeds step by step to progressively reveal its issues: their parts, their divisions, and sometimes their difficulties. Medieval logic issues were expressed in small manuals called *Parva logicalia* and there was a thirteenth century author, Peter of Spain, whose manuals, called *Tractatus* [1], were a point of departure for many authors of logic texts until the seventeenth century. At the beginning of the sixteenth century we find something very curious: the visual expression, that is, the expression of the content of Peter´s manuals through images in the *Logica memorativa* [2] of Thomas Murner (1475–1637).

Analogy has been described as a relation involving similarity and difference between two things and as a ratio between two pairs of things. There are at least two kinds of analogy: analogy of attribution and analogy of proportionality. Both have already been dealt with by Aristotle, who includes a treatment of metaphor in several of his writings. Metaphor and analogy are recurrent both in everyday speech and in written discourse. It is a fact that most of the time we understand metaphors, as if we had previously agreed with an original sense of one term and then grasp another sense of that term to understand a metaphor where it appears. Sometimes we do not even realize that we are dealing with metaphors, as it is often the case when reading fiction. In this contribution we want to show some visual images of logic matters emphasizing their analogical and metaphorical content.

## 2. Teaching Logic and Medieval Logic

The logic courses we teach our students at the baccalaureate and undergraduate level usually contain these topics: logic of propositions and the logic of quantifiers plus identity. Regarding the former we cover, among other things, these points: atomic and molecular propositions, their formalization, their connectives and truth tables, methods to establish which propositions are tautologies,

contradictions and contingent propositions. We also teach the students methods to establish the validity of argumentation and how to detect fallacies, resorting to a formal language specially designed for these tasks. In advanced graduate courses we can find the teaching of modal and non-classical logics. Regarding the later, the logic of quantifiers, predicate logic covers quantification sometimes appealing to the famous Square of Opposition, so popular in the Middle Ages.

Medieval logic tracts usually begin with the notions of "name", "verb", or "proposition", the verbal and conventional language. Medieval logic deals with several issues: terms, propositions and argumentation, which are also related to the operations and products of the intellect, so logic deals with the idea or concept, judgment, and reasoning. From this point of view logic may be regarded as a *scientia sermocinalis* and as *scientia rationalis* respectively. The study of terms leads to several divisions: categorematic and syncategorematic terms, univocal, equivocal and analogous terms, subject terms, and predicate terms. The study of propositions includes the categorical propositions and the hypothetical propositions, i.e., atomic and molecular sentences according to these days' terminology; it includes opposition and equivalence between categorical propositions that share subject and predicate in the same order, although their quantification may differ; it includes conversion of propositions that share the same subject and predicate, but not in the same order; it contains syllogistic and modal propositions; it comprises ascent and descent, which consists of transforming quantified propositions into chains of singular ones and vice versa; it embraces fallacies and their classification. It also deals with modal propositions and their relationships. The logic of quantifiers covers the Aristotelian syllogistic, including modal syllogistic, and it allows the logical treatment of more complex propositions involving relations and multiple quantification. We find not only squares, but also octagons of opposition to express these relationships among different kinds of sentences. We will find most of these topics in Murner's logic "flash cards".

Medieval thinkers called "imposition" (*impositio*) to that process by which a word is conventionally imposed to mean a thing. Well, the same process happens here, now applied to terms that already have a meaning to which another one is added; it is a kind of "second" imposition that does not abolish nor supersede the first. It is related to the doctrine of "intentions"; a first intention sign is that which refers to an extra mental and extra linguistic thing and a second intention sign refers to a sign. The *Trivium* (which includes grammar, rhetoric and logic) contains second intention terms such as these: "nouns", "verbs"; "genus", "species"; "metaphor", and "euphemism". They refer to a grammatical property, a logical and a rhetorical one respectively. The kind of "second" imposition I am referring to points out to a rather new use of words according to certain customs and uses of the speakers. Let me give a couple of examples from Peter of Spain. We can say in a conversation "the river runs" or "the meadow laughs" and we are understood by our listener because we are using secondary meanings. The words "it runs" and "it laughs" signify principally "to run" and "to laugh" and secondarily "to flow" and "to bloom". These last meanings come from use and custom, so we can say "the river runs" and "the meadow laughs" without any problem; language gets its force just by use, as Albert of Saxe [3] pointed out.[1] By the way, we may confuse the primary meaning with the secondary one committing thus a fallacy: everything that runs has feet, the river runs, therefore the river has feet. We are dealing with a metaphor here and the fallacy consists in extending too much such a metaphor. Murner's metaphors are of different kind, he uses images to convey meanings related to logic. They are metaphors at least in the sense that there is something (a visual image) to represent or to explain another thing (a logical content).

Murner tries to convey most of the themes above in his mnemotechnic images but we shall consider only a few, just enough to offer the reader a glimpse of his technique and of the variety of

---

[1]　" . . . quia hec verba 'currit' et 'ridet' per prius significant ridere vel currere et per posterius florere vel labi, quia hec significant ex propria impositione, illa vero ex assuetudine". Peter of Spain [1] (p. 101). " . . . sermo non habet vigorem nisi ex uso loquendi . . . ", Albert of Saxe [3] (p. 922, paragraph quoted).

objects used to transmit the logical content of the cards. Analogy plays an important role here, as we shall see.

### 3. *Logica memorativa* and Its First Image

The complete title reads *Logica memorativa. Chartiludium logice sive totius dialectice memoria et novus Petri hyspani textus emendatus: Cum iucundus pictasmatis exercitio: Eruditi viri.f.Thome Murner Argentini ordinis minorum: theologie doctoris eximij*. The book begins with a prologue, an exordium, a table indicating the symbols of each treatise and some instructions to manage the cards which are completed by further instructions at the end of the book, in a part labeled as "Exercitium". After this Murner explains the treatises using cards. There are 51 cards plus one which summarizes the content, so we do obtain 52 cards, as Jorge Medina points out [4] (p. xix). As a matter of fact, Murner is giving the students a little more difficult deck of cards.[2] Saint Exupery wrote a book for children, *Le Petit Prince* and when he dedicated it to an adult he had to rectify the dedication saying that it was for that same adult but when he was a child. The exordium of Murner's book is "to the studious youth" (*studiosa iuuentus*) and, as an apology for the book involving a set of cards, he emphasizes that it was written by someone in the prime of his youth (*in iuuentutis flore*). It contains sixteenth treatises each one beginning with a synthesis of one tract presumably from Peter of Spain, "presumably" because Peter's *Tractatus* contain only twelve treatises. Almost all of Murner´s treatises contain several chapters (*passus*); although we shall not review the differences between Murner and Peter we occasionally will say a few words about them.

The first image (Figure 1) we find in this book before the first treatise gives us some idea of what we will find in the logic cards. It is not a drawing by Murner, but a woodcut taken from *Margarita philosophica* [5] (p. 81) by Gregor Reisch (ca. 1467–1525).[3] It is a lady huntress who strides from left to right, well dressed and sounding a horn or trumpet from which a couple of roses come out. She is armed with a bow and wearing a curved sword in its sheath, leather shoes and slippers with metal tips. Ahead of her there are two greyhounds running after a hare which runs a little further, flanked by some flowers and little bushes. On the top left side there are four trees, three with some foliage and one without any leaves.

---

[2]　"Hec autem iactuum exercitatio acrior est et difficilior: chartiludiorum agitationi" says Murner in the "Exercitium". It is not easy to reconstruct how exactly was the game of these cards. Given the information supplied below, when dealing a card the student would had to recognize which treatise it belongs to, which chapter of that treatise is about, and its logical content.

[3]　Reisch's work deals with the *Trivium*, which comprises Grammar, Logic or Dialectics and Rhetoric, and with the *Quadrivium* but it is also related to natural and moral philosophy; as a matter of fact, it is an encyclopedia of the time. The reader will notice the differences between Reisch's and Murner's images in Holden [6].

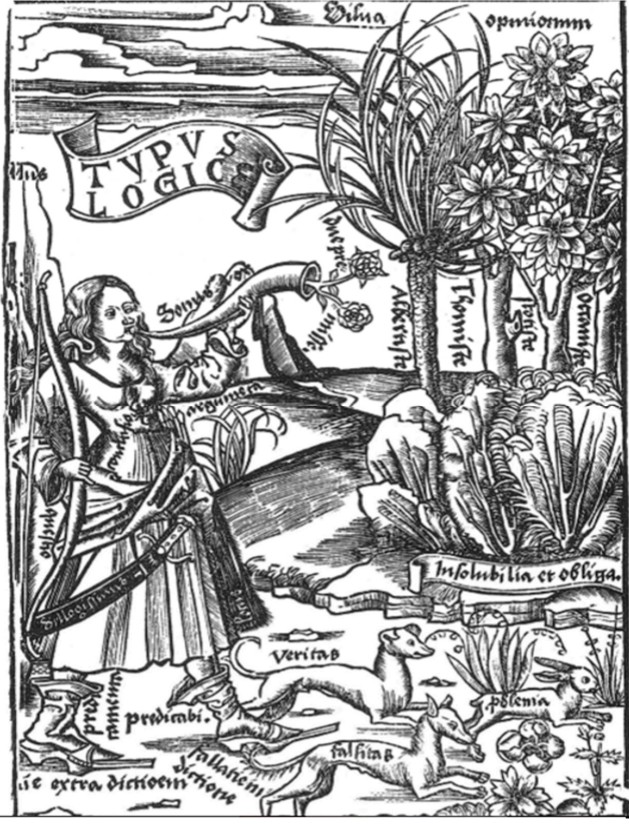

**Figure 1.** Typus logice.

What does this first image mean? The answer is not difficult to find since in addition to the image itself, there are words inside it that inform us about everything there; they are like labels imposed on things. It is not about sticking the word "sword" to the sword itself but the word "sillogismus" to the sword, or the word "questio" to the bow and "argumenta" to the arrows. The horn shows the words "sonus" et "vox" written above and next to each shoe we can read "predicamentum" and "praedicabile" respectively. Next to the trees we read "ockamiste", "scotiste", "thomiste", and "albertiste". In the right top we read "typus logice", a picture or portrait of Logic.

We can say, regarding Reisch's image, that we understand it because its meanings are glossed, so to speak; they can be literally read. It is an allegory where the lady represents Logic advancing towards or close to the forest of opinions among thorns and thistles, armed with dialectical weapons, her greyhounds are truth and falsehood and between them a "problem" represented by the hare that runs ahead of them. Thus, the syllogism and the *questio* are dialectical weapons (represented by the sword in its sheath and the bow), besides the shoes and arrows. The jungle of opinions is formed by different schools which take their names after their founders: Thomas Aquinas, William of Ockham, John Duns Scotus and, I guess, Albert of Saxony.[4] The thorns are the logical paradoxes (*insolubilia*) and "obligations" (*obligationes*), two of the most difficult logical issues in medieval logic; before them we find the fallacies of diction and extra-diction, probably overcome by the predicables and the predicaments as we read next to the woman's shoes. The reader will have no difficulty in recognizing what the image is about, since everything is already written there, and the reader of that time will have had a much better understanding than ours.

---

[4] Albert of Saxony, also spelled as Saxe, (ca. 1316–1390) is also known as Albertus Parvus or Albertucius to distinguish him from Albert the Great. His *Perutilis logica* and *Quaestiones in artem veterem*, constitute a good synthesis of medieval contributions already crystallized in the fourteenth century.

*3.1. Murner Cards and Treatises*

Murner's images are different, most of them do not contain words that explain their content, and when there are some words they are not explanatory even if they have some meaning or purpose. The reader will not find labels on the images the first time viewing the cards but will meet with numbers. Let me explain.

The book is divided into sixteen treatises, most of them containing several chapters and each chapter displaying an image containing the synthesis of the very theme dealt with there. The synthesis contains sentences or groups of sentences which are numbered. They describe, define, explain, divide and so on the addressed point. Then there is an image with numbers inside and finally we read Murner's explanation of the image according to those numbers. So, the numbers coincide: the enumerated exposition of the content, the numbers in the image and the explanation of the image according to the numbers. Unlike Reisch, Murner's images contain no explanatory words, but numbers each one indicating not a word but a sentence that explains the content of the image. There are some letters in card 26 with a mnemonic function, an illegible word in card 34, the word *quies* in card 38 and *in illo tempore* in card 49. These words are not used in the usual way. We shall see that the content is arbitrarily or conventionally proposed, just like a second imposition.

3.1.1. The Treatises' Contents, Symbols and Number of Cards

This list shows the treatises' theme, its symbol, the number of cards dedicated to that treatise. Each card shows several numbers and sometimes a number has two or more divisions (3a, 3b, and even 3f, for instance) but I do not list them here. There are 51 cards, all containing numbers. The card that has the highest number is that of treatise 3, chapter 7 and 12 numbers. Treatises 11 to 15 contain a single chapter, each one with two numbers. The cards of the other treatises range between numbers 3 and 11. The card of the first treatise contains 11 numbers.

1.  The enunciation, a jingle bell, eight cards
2.  The predicable, a lobster, six cards
3.  The predicament, a fish, eight cards
4.  The syllogism, an acorn, four cards
5.  The dialectical place, a scorpion, seven cards
6.  The fallacy, a female hat, eight cards
7.  The supposition, a heart, three cards
8.  The ampliation, a cricket, one card
9.  The restriction, a sun, one card
10.  The appellation, a star, one card
11.  The distribution, a bird, one card
12.  The exposition, a moon, one card
13.  The exclusion, a cat; one card for this and the next two
14.  The exception, a coat of arms
15.  Reduplication, a crown
16.  The descent, a coiling snake, one card

The first six correspond to Aristotelian logic and the remaining ten to the medieval contributions to logic. The reader can notice the amount of cards and numbers dedicated to Aristotle to the detriment of the medieval themes, which shows the Renaissance interest in Aristotle and the "return" to his logic.

We will find a symbol of each treatise in every card. If you take a card you will know to which treatise and chapter it refers to, for instance, if there are five lobsters you will know it corresponds to the fifth chapter of the second treatise (though there some "typos" in the first treatise).

### 3.1.2. The Wonder of the Images

If we take all the images of the cards, we notice something very complex and surprising. There are easily recognizable figures, both human and animal. Among the human ones we can find the king and the queen, the male and female servant, and also figures of people with a certain function or work: the woodcutter, the cook, the female cook, the bricklayer, the fisherman, the professor of Latin, the one-eyed man, the dyer, the spinner, the angel, the blind man, the woodcutter, the painter sketching Botticelli's *The Birth of Venus*, the plowman, the chess player, the potter demon, and many characters recognizable by the reader of the time. Among the animals we can find horses, cats, dogs, roosters, chickens, geese, pigeons, magpies, worms, snakes, fishes, and unicorns. There are many artifacts: crowns, hats, chess sets, blackboards, compass, brooms, ladders, hourglasses and mechanical clocks, plumb lines, lathes, strings, trumpets, scales, trawlers, saws, jars, broken pots, rulers, keys, locks, hooks, mirrors, boots, women's shoes, wheels, mortars, spears, swords, arrows, darts and their targets, a blazon, a monstrance, the cross of Saint Andrew, baskets of bread, and so many others, even a potty. But most important are their combinations. There are animals that look like elephants with a bird claw and bear leg, unicorns with deer horns, roosters with a scarf and a rosary on their paws, kittens tied to each other, fighting dogs, foxes, a demon with horse legs and tail, a demon with a halo of a saint, a child with hands bound, human and animal heads puffing smoke from their mouths, a queen on horseback showing in the one hand the severed head of Holofernes and in the other the Delilah's scissors, and so many other things that it is not possible to mention them all. At first sight they could seem grotesque or have no meaning at all but as we read Murner's explanations we understand things and keep them in our memory, because there are repetitions of the same figure in several cards, sometimes with some shift of meaning.

When Murner says that the magpie represents Dialectics we must accept what he says for otherwise we could not start the game of logic cards. When he says that Dialectics has to do with dialogue and discussion, we understand quite well when he shows two magpies arguing with each other. In some sense, we find a similar process as when we start reading fiction since we do not question what the author says. It is a kind of deal between Murner and the reader, a pact which makes us accept Murner symbology without asking him why. Murner himself affirms in card 38 that he strives to help memory, not to supply reasons.[5] Nevertheless, he is not completely arbitrary when proposing his images and symbols. By the way, looking at the magpies we cannot but remember those crows from the 2nd Century croaking on the rooftops "about what implications are sound".[6] Sometimes the dispute between two arguers may seem as a croaking debate, and even more, for Murner depicts in that very card a quarreling dog (*cane rixante*) because in an argument people concede and deny. Sometimes people say that an argument is a war [8] (p. 3) and Murner himself was acquainted with bitter arguments involving satires and even songs and pamphlets related to Luther and the Reformation, but that is another story (see [9]).

### 3.2. The First Card: The King, the Utterance

I ask the reader to look at the card below and try to guess its meaning, or meanings since we already know there are eleven from (Section 3.1.1) above and that the bell indicates the first treatise dedicated to enunciation. The bell appears several times in Image 2 (Figure 2): coming out of the mouth of the king, in both ears of the horse, in his rump; it also appears coming out of the mouth of the dog and the man in the horse's tack. The bell suggests sound, so we can guess that it refers to the sound that comes out of the mouth.

---

5　" . . . laboamus enim in adiutorio memorie: non in ratio dum datione". Treatise 6. Ch. 1. card 34. He adds he uses woman figures for a better remembering.
6　Cfr. I. M. Bochenski [7] (p. 116), the saying comes from Callimachus.

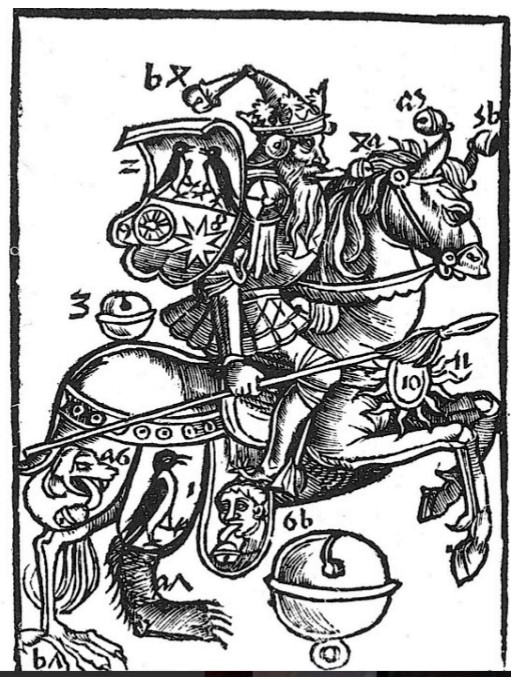

**Figure 2.** Card 1.

The magpie represents dialectics, according to Murner, because dialectic involves discussion, which occurs between two people, that is why there are two magpies. The bell in the king's tongue means the sound coming out of the mouth to form the voice. The bell in the cap represents a sound that is not a voice at all. In the right ear of the horse the bell represents the meaningful voice that is captured by the ear. In the left ear we find a broken bell to show the non-meaningful voice like: "buba", "blitiri". The dog barking (or the bell coming out from the dog's mouth) means the meaningful voice by nature (the moaning or crying mean naturally some pain) contrasting to the human face sticking out the tongue which represents the human voice, human language that is meaningful by convention. The voice (i.e., utterances) can be simple and complex, and it is shown by the hind legs of the horse; the right one is thinner and bony, it is a bird like leg while the right leg is fat and with muscle and hair, a bear like leg. This is to show the difference between the simple voice, which is "thinner", and the complex voice, which is "thicker". His examples: "man" and "man is an animal" respectively.

Below the magpies we find the North Star, it represents the name, whose meaning is stable. The wheel, on the contrary, represents the verb, because the verb involves movement, change; so, we have subject-term and predicate-term. They are separated in this image but naturally may make a sentence, as we shall see below.

The last numbers, ten and eleven, represent something rather unexpected: first and second intentions. It is unexpected because second intentions are not to be found in Peter of Spain's *Tractatus*, but no doubt it was a common doctrine at Murner's time. Number ten is in a mirror which stands for knowledge in the mind for first intentions reflect things as they are "contained" in mind. The rays are second intentions at number eleven. Let me put examples of sentences of first and second intention. In the sentence "Peter is an animal", "animal" is a first intention term since Peter is known as an animal but in the sentence "Animal is a genre", the term "genre" is taken as a second intention term. The second intention correspond to a kind of reference treated in the Supposition theory, called *suppositio simplex*, as exposed in card 44, the treatise on supposition. A second intention reflects a further activity of the mind.

3.2.1. Some Shifts of Meaning

Both subject and predicate terms are terms, and Murner takes advantage of this combining the North Star and the wheel to represent quantified subject terms and singular terms in card 2. I will call this combination the "North-wheel". Image 3 (Figure 3) shows it.

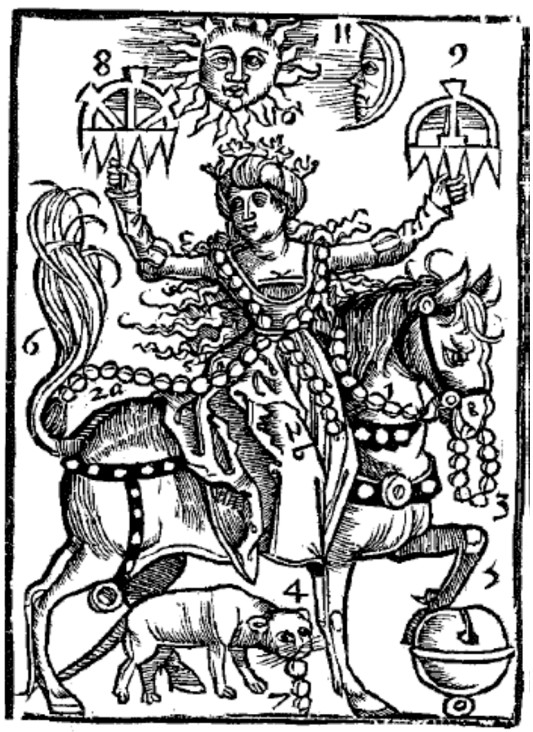

**Figure 3.** Card 2.

When the North-wheel has three spokes, it is to represent a sentence with a common noun as a subject: it may be universally or particularly quantified or having no quantifier at all (the Latin example *homo currit* exhibits no quantifier at all and this kind of sentence is called "indefinite sentence"), a one spoke North-wheel represents a singular term sentence like "Peter runs" for instance.

The sun and the moon stand for the universal and the particular quantifier, respectively. In the same card 2 Murner establishes the horse's left front leg as the subject of a sentence, since the subject goes before the predicate. The horse´s tail represents the predicate, since the predicate is to be placed after the subject. We obtain a categorical sentence. The four bells from the cub's mouth stand for universal, particular, indefinite and singular sentences. There is also a sort of play on words: the letters of "cat" are also the first letters of "catulus" (the cub) and "categorical". It helps the student to remember more easily.

Card 16 from treatise 3, second chapter shows the North-wheel assuming the role of the first and second substance. This parallels the distinction singular/common terms for the first substance always refers to an individual thing and the second substance to predicates. The second substances are to be said of the first ones. From here we can move to the notions of subject and predicate, which are found in card 23, Chapter 1 of treatise 4. Murner returns to the North-wheel to stand for singular terms in card 27, chapter 1 of treatise 5. He takes terms to be able to refer to themselves as linguistic entities in card 43, chapter 2 of treatise 7. This is his example: "Peter is a name" (*Petrus est nomen*) and terms also may be quantified, as they are in this sentence "every word 'man' is a monosyllabic word" (*omne ly homo est dictio dissilaba*). He also allows them to represent something else, as this example shows: "Socrates is an individual" (*Sortes est individuum*) but it may refer to the thing itself as in "Socrates runs" (*Sortes currit*) in card 44, chapter 3 of treatise 7. Murner uses the same "symbol", the North-wheel, to refer to different things. His combination of the North Star and the wheel makes it possible to refer

to several things at once. We can summarize as well: terms may refer to things, to concepts related to things and to terms themselves.

### 3.3. The Square of Opposition

We already know some things to understand the square´s image, card 4 from treatise 1, chapter four. We know that the horse's leg stands for the subject term; the sun stands for the universal quantifier and the moon stands for the particular one. In terms of quantity, the sun seems to illuminate more than the moon, and that is why it represents the universal quantifier. There is a Spanish saying that says "the sun rises for everyone" perhaps based on Matthew 5: 44–45. Anyway, Murner is a monk and that's a good reason to expect biblical references and sources for his teaching. Among other biblical references, card 48, treatise 11, refers directly to the Gospel account of the distribution of the loaves, Mark 6, 34–44. "Distribution" is a technical term for a movement involving a universal quantifier, which distributes the common term among its "inferiors" as everyone in the gospel story enjoyed the "distributed" loaf. The card shows five loaves inside the basket.

There is an almost natural convention according to which we classify things according to their quantity and where we spatially place them. Thus, the greater quantity is placed above and the smaller one below [8] (p. 15). This also happens in the square of opposition, so we will have the sun above and the moon below, or rather, two suns above and two moons below to incorporate their negative counterparts, i.e., the universal negative and the particular negative propositions. Murner had established, in card 3, a cord to represent negation when it is placed inside a sentence, as in this sentence "A man is not an animal" (*homo non est animal*). A cord because when someone is going to be punished standing before the cord he denies everything. The affirmative sentence is represented by a weight, which tends to go down by its own weight. He does not use any cord nor weight to represent negation and affirmation in his picture of the square of opposition. Perhaps it would be odd to place a weight in the superior part of the image, just next to the sun; perhaps he wants to deny the quantifier, not a sentence. He also uses the cord to negate terms, as in card 5 where the cord tying the leg and the horse's tail indicates conversion by contraposition. Anyway, Murner now uses a pair of objects designed to decorate the head but with a strong symbolic content: the heavenly diadem and the mundane garland. The celestial diadem represents the truth and the mundane the falsehood. A heavenly crown is to symbolize that there is true life in heaven while the garland represents falsehood since world's life is false and deceptive.[7]

We find the horse´s legs close to the sun and to the moon, this suggests that the quantified term is the subject term. There is no horse's tail at all, in case there were one we would have to explicitly quantify the predicate. Nevertheless, in card 23 there are two suns, one of them "shining on the predicate" (*cum cauda qua supra predicatum*), i.e., the horse's tail. It does not mean quantifying over predicates, it rather refers to the *dici de omni et de nullo*, predicating of everyone and of no one; there is a shift of meaning here, from "predicate", which is a noun, to "predicating", a verbal form. In card 24, the chapter on syllogism, the sun appears again indicating the need of at least one universal premise. In card 5 the sun is so close to the moon that it almost becomes it, this is to show the accidental conversion by which the universal quantifier becomes particular.

Once we know how to find quantifiers, subjects, truth and falsehood, it takes a little time to understand the card of the square (Figure 4). The upper part shows two suns, a diadem and a garland, so we have two universal sentences regarding the subject, one affirmative and one negative sentence. As the diadem in the upper central part comes first, the other universal sentence is to be a garland, since contrary sentences cannot be true simultaneously. The opposite holds for subcontrary sentences: if the first is a garland (false), the other must be a diadem (true), as shown in the middle lower part.

---

[7] "Celesti diademate verum significantur quod vera illic sit vita, serta falsum quod mundi vita falsa sit el fallax", Treatise 1, chap. 4, card 4.

There is a diagonal cross at the center of the image joining diadems with garlands, meaning that if one is true, the opposite contradictory is to be false. Two diadems at the left side represent subalterns being simultaneously true and two garlands at the right side represent subalterns being simultaneously false.

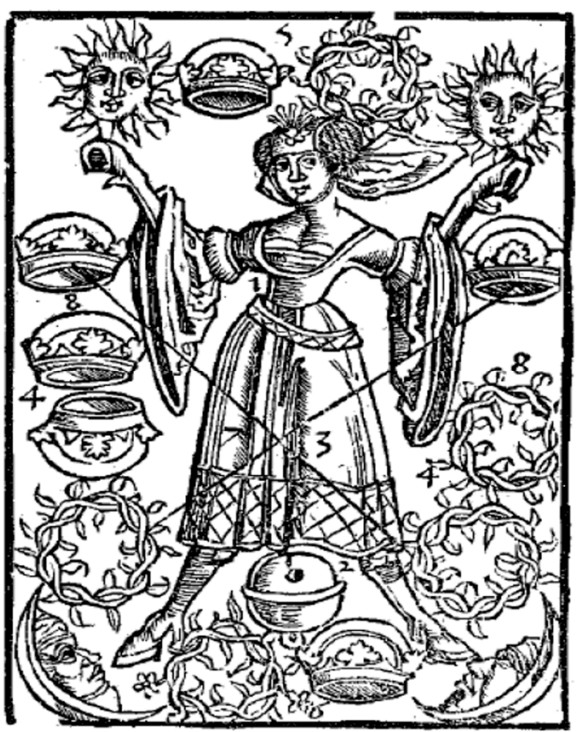

**Figure 4.** The Square of Opposition.

### 3.4. Card 14: Univocal and Equivocal Predication

There is no treatment of analogy in Murner's book nor in Peter's one. Analogy is to be found in some classification of terms by which we obtain analogous, equivocal and univocal terms. Murner is not treating terms, but predication, so what we do find here is a treatment of univocal and equivocal predication, which is quite natural since most of his book is dedicated to Aristotelian themes, beginning with themes from *Categories* and *On Interpretation*. Aristotle treats analogy in several places of his *Poetics*, *Rhetoric* and *Metaphysics*.

We understand what a man is, we have a concept of it and we also have a word for that concept, the word "man". When we recognize a man, we can say "this is a man" and we may predicate that concept to many individuals: "Plato is a man", "Socrates is a man" and so on. This is the so-called univocal predication. We may have a higher concept and predicate it to its inferiors: "man is an animal", "lion is an animal" and so on. This is also a univocal predication. Equivocal predication, on the other hand, preserves the word but not the concept and there could be several concepts under the same word, as it happens with the Latin word *canis*, which "is predicated equivocally to the barking dog, the sea fish and to the sky constellation".[8]

How does Murner represent this kind of predication? Predicating is a kind of verbal discourse, an act realized just by speaking and Murner has already used bells, concatenated bells to express the voice coming out of the mouth.

Now it is air coming out of the mouth; at first sight it could be taken even as smoke or as fire. Murner uses the word *anhelitus*, "breath" so one breath, one voice signifies one concept, one sense; many breathes at once signify many concepts, the so-called equivocal predication. Card 14 (Figure 5)

---

8　　"vt canis predicatur equiuoce de cane latrabili: pisce marino: et sidere celesti...", Treatise 2, chap. 6, card 15.

shows a man exhaling three voices because there are three meanings conveyed by one word, the word *canis*, but another head is exhaling just one breath, the univocal predication showed by a head at the man's hand. Looking at the card we recognize the treatise and the chapter by the lobsters: six lobsters means chapter six of the second treatise, on predicables. Card 15 shows univocal predication, one sense being represented by one breath coming out of the dog's mouth and when treating fallacies in card 36 Murner uses the same dog to represent the fallacy of equivocation, three breaths at once, i.e., when the same word is used to convey three different senses.

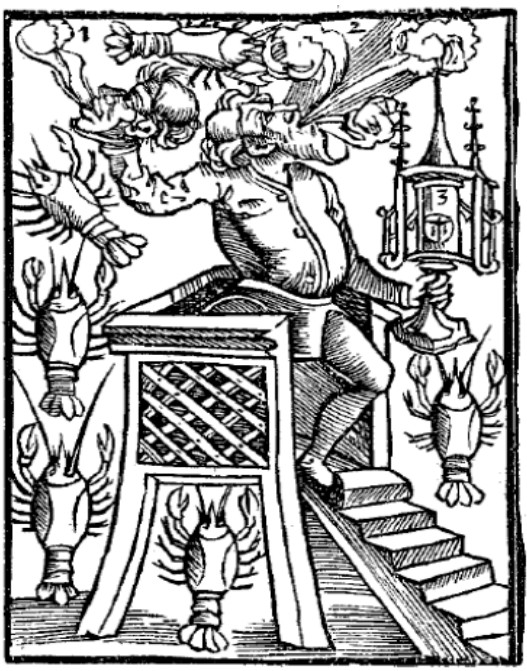

**Figure 5.** Card 14.

## 4. Some Concluding Remarks

It is not necessary to examine all the cards. I think we have found some clues about how Murner works. First, the objects are familiar: animals, utensils, characters (the king, the queen, the servant, etc.). Second, spatial placement is also important: forward, back, up, down. It has served to express the subject and predicate of sentences and universal and particular quantifiers. Third, there are some shifts of meaning, like the use of bells. They serve to express several things: sound, voice, simple and complex sentences, and even non-significative voice sounds, as the broken bell is intended to show. The first card combines animals: magpies, the dog, the human, and the horse with different legs. There is a sun with rays, the North Star, the wheel and the bells are everywhere. Certainly, the imaginative reader would be able to hear all the sounds suggested by the images.

Of course, the element of surprise must not be absent. The first image, the hunting maiden, is full of words, but the following ones do not contain any word (except for some cards). Instead of words we have numbers. We know that each number has a meaning and the reader, who will begin to find familiar figures, can guess the meaning of what he sees, or consult the text and know the meaning of it at once. There is a surprise when we discover, in card 8, a demon with a halo of holiness. This is impossible, and we read in the text that we are dealing with the notion of impossibility, expressed with that figure.

The word "analogy" does not appear in Murner's book, but the notion is present throughout the text. The similarities, for example, between the sun that shines on everyone and the universal quantifier, or the moon that illuminates some and the particular quantifier, are present in the text. That is why it is possible to use the figures of the sun and the moon as quantifiers. Perhaps it is better

to say "proportions", just as it is impossible for a demon to be holy, so an impossible sentence cannot be true. Just as the sun shines for all, the universal quantifier is distributed among all its subordinates.

With this I say goodbye to the reader. My intention has been to show Murner's book and the charm of his cards, and to suggest his relationship with analogy.

**Funding:** This research was funded by the Meritorious Autonomous University of Puebla, grant CABJ-EDH17-I.

**Conflicts of Interest:** The author declares no conflict of interest.

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
