# Peer review of "Analogy and Visual Content: The Logica memorativa of Thomas Murner"

_philosophies, doi:10.3390/philosophies4010002_

Round 1

Reviewer 1 Report

The paper presents Thomas Murner's "Logica memorativa" and shows some very interesting aspects of implicit uses of analogy. The Autor in a very clear way explain the mnemonic strategies for students in order to better and faster understand, for instance, the square of oppositions. The paper is very informative, original and opens further lines of research.

I would suggest only some minor corrections:

There are some unnecessary spaces: 73, 84, 101, 116, 118, 179, 198, 199, 219, 236, 247, 338, 348, 357.

47:  "language" instead of "languaje"

83-85: The title would be better in italics (in general, all book titles in italics, for instance l. 91)), "minorum" instead of "minoris" and please add space before "theologie".

In general, please put all captions rather under the images (Image 1, add space)

Footnote 3, should be "Reisch's work"

178: maybe the title of the painting (The birth of Venus") also in italics

185: "are their combinations"

There seem to be an editing problem with the third image

Author Response

Thank you very much for your corrections, I have emended the paper according to all of them. I have also corrected image 3 and added some lines for a better understanding of the decks.

Reviewer 2 Report

The book described, by Thomas Murner, seems very interesting. for my part you failed to explain the significance of the elements of the images adequately. (It's unclear why you call them decks, which is a work for a whole pack of cards. Are they not just cards?) 

Author Response

 Thank you for your comments. I have added a few lines to explain why I call "decks" to the cards. 

I have emmended some English typos and sintax construcctions. I have also added some lines about metaphor and the visual content of the images. All of them are written in blue characters. 

Round 2

Reviewer 2 Report

Unfortunately, the paper is still unclear, partly through the poor standard of English. It needs a complete rewrite, and only when that is done, thorough checking by a native speaker. 

Author Response

The paper has been revised by a native speaker.